# Sensorimotor Rhythm-Based Brain–Computer Interfaces for Motor Tasks Used in Hand Upper Extremity Rehabilitation after Stroke: A Systematic Review

**DOI:** 10.3390/brainsci13010056

**Published:** 2022-12-28

**Authors:** Jianghong Fu, Shugeng Chen, Jie Jia

**Affiliations:** 1Department of Rehabilitation Medicine, Huashan Hospital, Fudan University, Shanghai 200040, China; 2National Clinical Research Center for Aging and Medicine, Huashan Hospital, Fudan University, Shanghai 200040, China; 3National Center for Neurological Disorders, Shanghai 200040, China

**Keywords:** brain–computer interfaces, motor task, sensorimotor rhythm, stroke, hand rehabilitation

## Abstract

Brain–computer interfaces (BCIs) are becoming more popular in the neurological rehabilitation field, and sensorimotor rhythm (SMR) is a type of brain oscillation rhythm that can be captured and analyzed in BCIs. Previous reviews have testified to the efficacy of the BCIs, but seldom have they discussed the motor task adopted in BCIs experiments in detail, as well as whether the feedback is suitable for them. We focused on the motor tasks adopted in SMR-based BCIs, as well as the corresponding feedback, and searched articles in PubMed, Embase, Cochrane library, Web of Science, and Scopus and found 442 articles. After a series of screenings, 15 randomized controlled studies were eligible for analysis. We found motor imagery (MI) or motor attempt (MA) are common experimental paradigms in EEG-based BCIs trials. Imagining/attempting to grasp and extend the fingers is the most common, and there were multi-joint movements, including wrist, elbow, and shoulder. There were various types of feedback in MI or MA tasks for hand grasping and extension. Proprioception was used more frequently in a variety of forms. Orthosis, robot, exoskeleton, and functional electrical stimulation can assist the paretic limb movement, and visual feedback can be used as primary feedback or combined forms. However, during the recovery process, there are many bottleneck problems for hand recovery, such as flaccid paralysis or opening the fingers. In practice, we should mainly focus on patients’ difficulties, and design one or more motor tasks for patients, with the assistance of the robot, FES, or other combined feedback, to help them to complete a grasp, finger extension, thumb opposition, or other motion. Future research should focus on neurophysiological changes and functional improvements and further elaboration on the changes in neurophysiology during the recovery of motor function.

## 1. Introduction

Stroke causes the highest morbidity associated with disability-adjusted life years lost in China, with two million new cases annually [1]. Up to 66% of stroke survivors experience upper limb and hand motor impairments, which results in functional limitations in activities of daily living and decreased life quality [2,3], and leads to a heavy burden for the family and society. Hand rehabilitation after a stroke is difficult during neurorehabilitation. Traditional rehabilitation methods cannot fully meet the need of patients and the expectations of doctors [4]. Various methods were being applied in hand function rehabilitation, including central interventions such as mirror therapy, transcranial magnetic stimulation, transcranial direct current stimulation, brain–computer interfaces (BCIs), motor imagery, etc., peripheral interventions such as a robot, physical therapy, functional electrical stimulation, etc., and medicine such as botulinum for spasticity [5]. BCIs have been proven to be effective for hand motor recovery after stroke [6,7,8]. According to its working mechanism, BCIs can be classified as assistive or rehabilitative devices based on their clinical applications. In some laboratories, assistive BCIs are used as communication tools for amyotrophic lateral sclerosis patients [9,10] or as daily activity assistance for tetraplegia, such as drinking assistance [11,12]. Meanwhile, rehabilitative BCIs are mainly used in promoting functional recovery for such as stroke patients.

There were various kinds of rehabilitative BCIs equipment. In practice, the workflow of BCI is acquiring brain signals, extracting features, transforming the signal into command via external devices, and activating the sensory feedback. In non-invasive systems, BCIs involve brain activities measured by different kinds of equipment, such as electroencephalograph (EEG), functional magnetic resonance imaging, and functional near-infrared imaging, and the user’s movement intention such as motor imagery or motor attempt is decoded in real-time from the ongoing electrical activity of the brain by extracting relevant features [6]. Based on different features, such as common spatial pattern (CSP), and event-related desynchronization (ERD), different movement intentions can be classified by linear discriminant analysis (LDA) classifier, support vector machine. The algorithm converts the brain signals into information, then the external devices, such as the computer screen, robot, functional electrical stimulation (FES), or orthosis received the information and provide feedback to the subjects. The whole process forms a closed loop called neural feedback [13,14]. Auditory, visual, tactile, and proprioceptive feedback is commonly adopted in BCI, and their combination is used extensively in clinical experiments [15,16,17]. Motor imagery (MI), motor attempt (MA), or motor execution (ME) can activate several signal rhythm changes in the cerebral cortex [18,19], which can be captured and used to modulate the amplitude of sensorimotor rhythm (SMR) to control external devices. In addition, BCIs are a kind of active rehabilitation device. They achieved control of devices by catching the subject’s initiative. In particular, the motor task is not only the start factor for BCIs, but also the repeated training of task can promote motor recovery. Well-designed motor tasks and befitting feedback for the patients can enhance the BCI training and lead to a successful rehabilitation process.

Motor tasks usually concern the movement intention or the actual movement of the paretic limb. As we all know, hand recovery is a long and rough process. Several stroke patients in the acute stage can hardly move their hands, neither completely nor incompletely grasp, and they encounter kinds of difficulties, such as opening the fingers or moving their thumb or other fingers independently. The recovery of the hand function conforms to some rules, such as the six Brunnstrom recovery stages [20], but many of the hand functions stagnate at some specific stages. According to Brunnstrom recovery stage for stroke, in stage Ⅰ, there is no muscle contraction at all; in stage Ⅱ, there is subtle flexion of the hand; in stage Ⅲ, the hand can flex more actively but cannot be opened; in stage Ⅳ, patients can volitionally extend the thumb and other fingers partially; in the stage Ⅴ, patients can hold a ball or a cylinder, and they can extend their fingers simultaneously; and in the stage Ⅵ, the paretic hand can almost accomplish every kind of functional grasping and extending, but the speed and coordination are a little bit worse than the contralateral limb. The recovery rules can also be applied to shoulders, elbows, forearms, and wrists. It is obvious that an improvement from no active movement to active movement is a hard step, and the separation movement, from finger grasping to finger opening, is also a difficult process. Therefore, facing a series of difficulties, the task specificity of hand motions is of great importance, and the correlation with feedback is the main link and is discussed in detail in the review.

Clinical efficacy in hand function rehabilitation of stroke patients has been revealed by several reviews. Remsik et al. [21] considered BCIs as a method of hand function rehabilitation after stroke with a review. Monge–Pereira et al. [22] suggested EEG-based BCIs interventions may be a promising rehabilitation approach in subjects with stroke by a systematic review. Carvalho et al. [23] suggested that neurofeedback training with EEG-based BCIs might promote both clinical and neurophysiologic changes in stroke patients. Bai et al. [24] investigated the effectiveness of BCIs in restoring upper extremity function after stroke. Even though they have testified to the effectiveness of the BCIs, hardly have they forced deeply on these questions: Why was a motor task chosen in each BCI trial, and is it suitable for a stroke patient? Some stroke patients received BCIs training but gained small improvements. Except for other reasons such as a lack of treatment times, short treatment duration, etc., can the motor tasks help promote better motor recovery? Thus, this review concentrates on the motor tasks and feedback of BCI clinical trials based on upper limb and hand interventions with BCIs systems in patients after stroke, which is truly suitable for them to solve their problems. We use the traditional method to search articles and draw clinical recommendations. This review aims: (1) to explore the motor tasks design in EEG-based BCIs clinical trials, (2) to analyze the association between motor tasks and the neurologic mechanism, and (3) to discuss the feedback combing the motor tasks that were suitable for stroke patients.

## 2. Methods

### 2.1. Search Strategy

We searched articles in PubMed, Embase, Cochrane library, and Web of Science. At the same time, we screened the reference of previous systematic reviews in PubMed in case of missed articles. For PubMed, JF and SC took advantage of subject terms and entry terms for each subject, extended each subject term with the virtue of mesh categories, and then searched the corresponding entry terms separately. These subject terms and entry terms can be the reference for other databases. However, due to different search strategies in each database, other databases went through the same process to retrieve articles. The specific search strategy for each database can be found in Appendix A.

### 2.2. Study Selection

The inclusion criterion is following the PICO principle:(1)Subjects were hemiplegic paralysis, and were diagnosed with ischemic or hemorrhagic stroke;(2)Subjects received EEG-based BCIs training, which described the motor tasks in detail in the papers, and the control group received conventional therapy or sham BCI training;(3)The Fugl–Meyer Assessment Upper Extremity Scale (FMA-UE), Action Research Arm Test (ARAT), the Jebsen Hand Function Test, etc. were used for functional recovery assessments;(4)We concentrated on randomized controlled trials.

The PEDro scale was used to assess the methodological quality of the controlled studies (details are shown in Table 1).

## 3. Results

We searched articles in PubMed, Embase, Cochrane library, Web of Science, and Scopus and obtained a total of 442 articles. After screening, 15 randomized controlled studies were eligible for analysis. We mainly concentrated on the motor task design and the corresponding BCI system feedback in their research (characteristics of the enrolled studies are shown in Table 2).

### 3.1. Motor Task

Detailed information about different kinds of BCI motor tasks such as MI, MA, and ME is listed as follows.

#### 3.1.1. MI Task

The movement of the paretic hand is the main point in the design of the BCIs experiment. Imaging finger movements [25,36,37,38], including grasping alone, grasping, and opening [39,40,41], was applied in several experiments. Ang et al. [25] recruited stroke patients to receive hand grasp motor imagery training, and the control group received robot training. Finger extension imagery is still common in trials. Rayegani et al. [27] instructed the experimental group to contract the abductor pollicis brevis muscle and perform thumb opposition. Pichiorri et al. [28] assigned the MI task to imagine a sustained grasping movement and sustained complete extension of the finger. In Frolov et al.’s experiment [30], the experimental group received BCI-arm exoskeleton training, and the patients kinesthetically imagine a continuous opening of the right hand and the left hand.

Motor tasks may also involve movements of multiple joints, including the joints of the shoulder, elbow, and wrist. In several BCI research, patients were instructed to imagine the extension of the wrist [33,35]. Angand Chua et al. [26] instructed patients to imagine their paralyzed hands to reach out and reach the clock-face target on the computer screen. In another experiment [16], patients who were severely injured were enrolled and divided into two groups. The motor task was to imagine moving their affected hand toward the target indicated on the 8-point clock face on the computer screen. Stroke patients were recruited by Li et al. [15] to test the efficacy of MI-based BCIs training. Before treatment, the subjects in the BCI group were trained to complete MI tasks for effectively performing MI. They were trained to practice the experienced task, such as drinking water, and complete the MI task through a video by the unaffected hand. Then during the training course, the subjects were instructed to imagine the upper extremity movements according to the direction of the arrow following the voice “Begin to imagine left/right” and the randomized green arrow with a left or right command showing on the computer screen. In Cheng’s study [34], they designed some movements from activities of daily living (ADL), such as scanning goods, moving an object upward to a cabinet, using two hands to move a towel, pouring water into a cup, eating actions, and fine motor movement of picking up a small block using two fingers. The MI task was imagining arm movements and was matched to the performance of the ADL movements that were talked about above.

#### 3.1.2. MA Task

Attempting to extend fingers has also been applied in some experiments. Biasiucci et al. [7] asked the patients to attempt to extend or rest the affected hand (both fingers and wrist). Chen et al. [32] designed a study for subacute stroke patients, in which the BCIs group patients attempted to extend the wrist of the paretic hand. Ramos–Murguialday et al. [17] conducted a randomized controlled clinical trial, and trained patients to move the upper limb and reach forward with the help of arm orthosis. When the patients heard the corresponding auditory cue, they were instructed to try to reach, grasp, and bring an imaginary apple to their lap, and finger extension was involved in the reaching and grasping movement. In another study [31], patients were instructed to try to move their paretic upper limb to open and close the fingers or move forward and backward.

### 3.2. Different Feedback for the Motor Tasks

There are various types of feedback in MI or MA tasks for hand grasping and extension, including visual, auditory, tactile, and proprioceptive feedback.

#### 3.2.1. Visual

Visual feedback is a common choice for the BCIs experiment [16,27,28,34], and it is often presented as a computer screen or projection screen. In practice, patients watch the muscle contractions on the screen [27] as a game or some curtain displaying the simulated hand which demonstrated the imaginary movement [27,28]. In some combination cases, visual and movement feedback was provided by the Manus shoulder–elbow robot, and the exoskeleton assist the paretic arm moved from the center to the target displayed on the screen and back along a predetermined robotic trajectory [16,26].

#### 3.2.2. Proprioception

(1)Orthosis

The orthosis was used to assist the paretic limb to move. In some research, the paretic hand was attached to the orthosis to drive fingers extending (hand opening), and other researchers used arm orthosis (reaching) to assist the upper arm extension. The arm and hand orthoses targeted the patient’s ability to open and close the hand [17]. In another study [31], robotic orthosis was used to open and close fingers or move the paretic upper limb forward and backward just like the given motor task. The level of paresis determined the kind of movement to be performed during BCIs training, but all patients performed the movement of opening and closing the fingers. When the mu ERD was detected after the cue instruction to imagine finger extension, the star-shaped cursor moved down on the screen as visual feedback, and then the motor-driven orthosis extended their affected fingers [42].

(2)Robot

The robot used in BCIs is usually in the active-assist mode. The active assist mode likely generates greater proprioceptive sensory signals to the brain than the active non-assist mode does [43]. The Haptic Knob robot helped with the hand-grasping action [25]. They carried out an MI-based BCIs and tactile selective attention experiment. In the MI group, kinesthetic motor imaging (KMI) of the left or right hand was performed according to the direction of the arrow presented on the screen. In the tactile selective attention group, vibration stimulation of the left and right thumbs was implemented. Some KMI movements may be designed with ADL movements, and their feedback from the soft robotic glove is moving the fingers [34].

(3)Exoskeleton

The exoskeleton was also applied in a BCI designed to produce proprioceptive feedback. Frolov et al. [30] instructed patients to imagine the extension of their left or right hand, and the exoskeleton helped them to extend their fingers after receiving the commands. After the BCIs system correctly recognized the intention of the patient’s motor attempt, it would output command and manipulate the exoskeleton, driving the patient’s affected hands to complete the wrist extension motion [32].

(4)FES

The FES can also be used as feedback in the BCIs system. When patients correctly imagined the movement and their attention level went up the attention threshold, FES was triggered and stimulated wrist extensor muscles of the affected upper extremity. If a “movement attempt” was detected, FES would be triggered, with which a single bipolar channel was applied on the affected limb to inject current into the extensor digitorum communis muscle [7], and the threshold that initiated FES was adjusted after each run for each patient by the therapist, to determine the task difficulty. The FES can act on any muscle as requested, such as extensor carpus radialis muscles [15] or extensor carpi ulnaris [35].

## 4. Discussion

We summarized the commonly used MI/MA tasks in the BCI experiments. MI [44] refers to mental activity that involves specific movements without actual movement. An example of kinesthetic motor imaging involves imagining the feeling of the hand opening from the perspective of the first person while maintaining muscle relaxation. MA [45] is attempting to move the paralyzed limb while there is still no actual or little movement, and the electromyography (EMG) activities in the affected arm are several times higher during the motion phase than those in the rest phase. MI has been considered a therapy for promoting motor recovery after stroke [46], and they were often connected to the BCIs equipment. BCI has been proven to be effective in subacute and chronic stages of hand recovery of stroke patients. However, patients might encounter kinds of hand recovery difficulties during their rehabilitation courses. In the literature we have referred to, for the MI task, hand-grasping imagery [25,36,37,38], involving grasping alone, grasping, and opening [39,40,41], was applied in several BCIs studies. Finger extension imagery is also common in the BCI trials [27,28,30,42]. In some research, patients were instructed to imagine the extension movement of the wrist [33,35]. Grasping and opening are basic functions of a hand, and many motor tasks were designed based on these motions. At the same time, motor tasks may also involve movements of multiple joints, including the joints of the shoulder, elbow, and wrist [15,16,26,34]. As for the motor attempt task, attempting to extend fingers has also been carried out in several experiments [7,17,31,32].

There are various types of feedback in MI or MA tasks for hand grasping and extension, including visual, auditory, tactile, and proprioceptive feedback. Many experiments are designed from their combinations. Proprioception was more frequently used and in a variety of forms, including orthosis [17,25,31,42], robot [16,25,26], exoskeleton [30,32], and functional electrical stimulation (FES) [7,15,29]. We analyzed the specific motor task adopted in BCI experiments. Some researchers trained patients on the MI ability before the treatment to obtain good training effects, some may increase the threshold that initiates the FES to enhance task difficulties, and other experiments set two tasks for hand and arm with the help of a specific robotic orthosis.

As is known to all, motor recovery and functional improvements mainly depend on motor training, and task-based motor relearning is also important for hand rehabilitation after a stroke. After repetitive motor training, the motor function improved following the brain plasticity. Although motor tasks are an essential and non-negligible part of a BCI system, how to choose a motor task of the BCI training system for a stroke patient with hand motor dysfunction remains an unsolved problem, and is of great importance to their clinical outcome. MI and MA tasks are common experimental paradigms in EEG-based BCIs trials. They were used in post-stroke hand function rehabilitation. Patients who received BCIs training were asked to perform motor tasks of different motions. These motions included grasping, fingers extending, and wrist extending. All these motor tasks designed are essential for rehabilitative BCIs. From a meta-analysis published recently [24], we have known that MA in BCIs training appears to be more effective than MI, and we believe that MA may be a better choice in BCIs trials, especially since the task can be referred to the functional status in stroke patients. However, for people with different levels of hand motor impairments, what kind of motor task should be designed for them? As has been mentioned above, the Brunnstrom stage recovery rule for stroke can be an indication for study design. During the recovery, the process from no to mild active movement and segregation movement is a difficult step, so setting the proper task is an urgent need. In Ramos–Murguialday et al. [31] research, they designed two motor tasks for different levels of paresis in stroke patients. Thus, according to the rules, we may mainly focus on patients’ difficulties. For patients with no actual movement, we may focus on basic primary functions, such as grasp ability, with the help of embodied BCI feedback, to practice the motion repeatedly.

Different motor tasks may also match different types of feedback. The proprioceptive feedback is provided by an exoskeleton, orthosis, robots, or FES. In addition, virtual feedback can be another type of feedback that directly enters the brain. Feedback is a key element in motor rehabilitation in clinical work, and it has been reported to enhance brain plasticity and promote neural remodeling after rehabilitation training. Thus, varying modalities of feedback have been employed during BCI training. A combination of two or more feedback may create an enriched multi-element environment and be more helpful for stroke rehabilitation. Some studies have demonstrated that proprioceptive feedback is more suitable than visual feedback for entraining the motor network architecture during the interplay between motor imagery and feedback processing [47], and thus, it results in better volitional control of regional brain activity, but the two above are often combined in practice. In Bai et al.’s meta-analysis [24], they made a subgroup analysis, focusing on the relationship between different feedback and effectiveness. They mainly analyzed robot, FES, and visual feedback, and the results indicated that only BCIs triggering the stimulation of FES had a significantly larger effect size on motor function recovery, compared with control interventions. In Xie’s meta-analysis [48], they considered BCI combined with FES or visual feedback may be a better combination for functional recovery than a robot.

However, up to now, the mechanism of BCI promotes motor recovery is not very clear. MI-based BCIs involve neural mechanisms that volitionally control the movement of the hand [21], guide nerve plasticity, and enhance the connection between the motor area and the ipsilesional hemisphere [6]. Some studies have shown that after BCI training, the sensorimotor cortex of the ipsilesional hemisphere participates more, which might increase the excitability of the ipsilesional hemisphere [49,50] and change the rhythm of EEG, such as producing stronger ERD [15,28]. Patients with better SMR control may have higher functional improvements [51], and the performance of BCI is related to the improvement of motor function [15,30,52,53]. The proprioceptive sensory signals from these movements reach the motor cortex, the activation or continuous sensory input to the motor cortex of the ipsilesional hemisphere [21,54,55], and increase the afferent feedback, which has been considered useful for improving motor learning [56,57]. The recruitment of muscle spindles and Golgi tendon organs via FES may be effective. Some researchers believed that FES depolarized more motor and sensory axons, sending larger sensory volleys from muscle spindles and Golgi tendon organs into the central nervous system [58], FES or tactile input accompanying MI can produce stronger ERD [59,60], and the monosynaptic excitatory projections from spindles onto motoneurons may activate them concurrently with the presumed descending cortical command, thereby causing Hebbian association.

Facing the current need of patients as well as physicians, we need to design suitable motor tasks and choose corresponding BCI feedback to reach a higher hand function recovery after stroke. If a patient has difficulty extending fingers, the motor task can be designed as an attempt to extend fingers. Then the FES or the robot should assist the patient to extend fingers in an active-assist mode. Similarly, if a patient has difficulty flexing fingers, the motor task can be designed as an attempt to flex fingers and the FES or the robot will also be used as assistance with various feedback. However, there are some limitations to this review. First, our review only focuses on RCT. Thus, more motor tasks could not be presented. Second, the effects of different motor tasks and different feedback were not quantified. Future research should focus on neurophysiological changes and functional improvements and further elaboration on the changes in neurophysiology during the recovery of motor function, which may promote the development of BCI in the neurological field fundamentally.

## 5. Conclusions

To sum up, we focus on the motor tasks adopted in EEG-based BCIs research, as well as the corresponding feedback adopted in the BCI trial from the very perspective of the clinic. Many motor tasks involve imagining or attempting to grasp or extend the hand and were matched with the BCIs-triggered robot or FES combined with visual feedback. To optimize BCI rehabilitation training, we should focus on patients’ difficulties during BCI training to help them to complete grasp motions, finger extension, thumb opposition, and other complex motions with the assistance of the robot or FES, or other combined feedback.

## Figures and Tables

**Table 1 brainsci-13-00056-t001:** Methodological quality assessment of the enrolled studies.

Author/PEDro Item	1	2	3	4	5	6	7	8	9	10	11	Total
Ramos–Murguialday et al., 2013a [17]	1	1	1	1	1	1		1		1	1	8
Angand Guan et al., 2014a [25]	1	1		1			1	1		1	1	6
Li et al., 2014a [15]	1	1	1	1				1	1	1	1	7
Angand Chua et al., 2014a [26]	1	1	1	1			1	1		1		6
Rayegani et al., 2014a [27]	1	1		1			1			1	1	5
Ang et al., 2015a [16]	1	1		1	1			1	1	1	1	7
Pichiorri et al., 2015a [28]	1	1	1	1				1		1	1	6
Kim et al., 2016 [29]	1	1	1	1			1	1		1	1	7
Frolov et al., 2017a [30]	1	1		1			1			1	1	5
Biasiucci et al., 2018a [7]	1	1	1	1	1		1	1	1	1	1	9
Ramos–Murguialday et al., 2019a [31]	1	1	1	1	1	1	1			1	1	8
Chen et al., 2020 [32]	1	1		1				1	1	1	1	6
Miao et al., 2020 [33]		1		1				1	1	1	1	5
Cheng et al., 2020 [34]	1	1		1			1	1	1	1	1	8
Chen et al., 2021 [35]	1	1		1				1	1	1	1	7

1. Eligibility criteria were specified; 2. subjects were randomly allocated to groups; 3. allocation was concealed; 4. the groups were similar at baseline regarding the most important prognostic indicators; 5. there was blinding of all subjects; 6. there was blinding of all therapists; 7. there was blinding of all assessors; 8. measures of at least one key outcome were obtained from more than 85% of the subjects initially allocated to groups; 9. all subjects for whom outcome measures were available received the treatment or control condition as allocated; 10. the results of between-group statistical comparisons are reported; 11. the study provides both point measures and measures of variability.

**Table 2 brainsci-13-00056-t002:** Characteristics of the enrolled studies.

Study, Year	Country	n(E/C), Study Design	Experimental (E)/Control Group (C)	Feedback	Outcome Measures	Dosage	Main Results
Ramos–Murguialday et al., 2013a [17]	Germany	16/16, RCT	E: PT rehab + BCI-orthosis MA task: attempt to open and close the hand and arm extension C: PT rehab + sham BCI	The hand orthosis drives extending fingers, and arm orthosis assists the upper arm extension.	FMA-UE, GAS, MAL, MAS	40 min/d, 5 d/wk, 4 wk, 20 d	FMA-UE scores improved more in the experimental than in the control group, FMA-UE scores (3.41 ± 0.563, *p* = 0.018).
Angand Guan et al., 2014a [25]	Singapore	6/8, RCT	E: mobilization + BCI-robot MI task: imagine hand grasping C: mobilization + robot	The haptic knob robot for the hand grasping action.	FMA-UE	1.5 h/d, 3 d/wk, 6 wk, 18 d	FMA-UE score improved in all groups, but no intergroup differences were found at any time point.
Li et al., 2014a [15]	China	7/7, RCT	E: Con-rehab + BCI-FES, visual and auditory feedback MI task: imagine the upper extremity movements according to the direction of the arrow C: Con-rehab + FES	Once patients correctly imagined the movement five times in succession, FES was triggered, which stimulated the affected upper extremity’s extensor carpus radialis muscles.	FMA-UE, ARAT, EEG	1–1.5 h/d, 3 d/wk, 24 d	A significant improvement in the motor function of the upper extremity for the BCI group was confirmed (*p* < 0.05 for ARAT), simultaneously with the activation of bilateral cerebral hemispheres.
Angand Chua et al., 2014a [26]	Singapore	11/14, RCT	E: BCI-Manus robot MI task: imagine moving the paretic arm and hand forward to reach for an imagery target in front of them and to reach the clock-face target C: Manus robot	passive resistance-free movement of the paretic arm within the exoskeletal arm from the center toward the target displayed on the screen, along with visual feedback.	FMA-UE	1.5 h/d, 3 d/wk, 4 wk, 12 d	No intergroup differences (*p* = 0.51).
Rayegani et al., 2014a [27]	Iran	10/10, RCT	E: con-rehab + BCI-visual feedback MI task: contract the abductor pollicis brevis muscle and perform thumb opposition C: Con-rehab	Patients were provided with visual and audio feedback by watching the contractions on the screen as a game (puzzle).	JHFT	30 min/d, 5 d/wk, 2 wk, 10 d	No intergroup differences.
Ang et al., 2015a [16]	Singapore	10/9, RCT	E: tDCS + BCI- robot MI task: imagine moving their stroke-affected hand toward the target indicated on the 8-point clock-face video game. C: sham tDCS + BCI- robot	Passive resistance-free movement of the paretic arm within the exoskeletal arm from the center toward the target is displayed on the screen, along with visual feedback.	FMA-UE	1 h/d, 5 d/wk, 2 wk	No intergroup differences. Online accuracies of the evaluation part from the tDCS group were significantly higher than those from the sham group.
Pichiorri et al., 2015a [28]	Italy	14/14, RCT	E: con-rehab + BCI-visual feedback MI task: imaging a sustained grasping movement and sustained complete extension of the finger. C: con-rehab+ MI	A simulated hand was projected to demonstrate the imaginary movement as the visual feedback.	FMA-UE, MAS, EEG	30 min/d, 3 d/wk, 4 wk. 12 d	The FMA-UE score improved (*p* < 0.03) in the BCI group.
Kim et al., 2016 [29]	USA Korea	15/15, RCT	E: con-rehab + AOT + BCI-FES ME: Participants performed 18 action observational tasks related to their daily living by watching DVDs of a sequence of movements that should be performed with their own hands including (1) folding a towel, (2) cutting a toilet roll, (3) using scissors, (4) tightening shoelaces, (5) opening and closing a square airtight container, (6) opening a bottle top, (7) turning a faucet, etc. C: con-rehab	If patients correctly imagined the movement and their attention level went up to the attention threshold, FES was triggered and stimulated wrist extensor muscles of the affected upper extremity.	FMA-UE, MAL, MBI	30 min/d, 5 d/wk, 4 wk, 20 d	The FMA-UE was significantly higher in the BCI-FES group (*p* < 0.05).
Frolov et al., 2017a [30]	Russia	55/19, RCT	E: Con-rehab + BCI-arm exoskeleton MI task: Kinesthetic imagination of a continuous opening of the right hand and the left hand. C: Con-rehab + sham BCI	The hand exoskeleton helped patients to extend their fingers.	FMA-UE, ARAT	30 min/d, 3 d/wk, 12 d	Both groups improved in FMA-UL. Upon training completion, 21.8% and 36.4% of the patients in the BCI group improved their ARAT and FMA-UE scores respectively.
Biasiucci et al., 2018a [7]	Switzerland	14/13, RCT	E: BCI-FES MA task: attempt to extend the affected hand, fingers, and wrist. C: Sham BCI	If a “movement attempt” was detected, FES was triggered, with which a single bipolar channel is applied on the affected limb to inject current into the extensor digitorum communis muscle.	FMA-UE, MRC, MAS, EEG	1 h/d, 2 d/wk, 5 wk	BCI patients exhibit a significant functional recovery after the intervention. EEG analysis pinpoints significant differences in favor of the BCI group, mainly consisting of an increase in FC between motor areas in the ipsilesional hemisphere.
Ramos–Murguialday et al., 2019a [31]	Germany	16/12, RCT	E: PT rehab + BCI-orthosis MA task: instructed to try to move their paretic upper limb. (1) open and close the fingers or (2) move the paretic upper limb forward and backward. C: PT rehab + Sham BCI	The robotic orthosis was used to open and close the fingers or move the paretic upper limb forward and backward just like the given motor task.	FMA-UE, GAS, MAL, MAS	1 h/d, 5 d/wk, 4 wk, 20 d	The experimental group presented with FMA-UE scores significantly higher in Post2 (13.44 ± 1.96) as compared with the Pre-session (11.16 ± 1.73; *p* = 0.015).
Chen et al., 2020 [32]	China	7/7, RCT	E: BCI + exoskeleton + co-rehab MA task: attempt motion of wrist extension C: co-rehab	The exoskeleton drives the patients’ affected hands to complete the wrist extension motion.	FMA-UE	40 min/d, 3 d/wk, 4 wk	Both the BCI group (*p* = 0.032) and the control group (*p* = 0.048) improved in FMA-UE scores.
Miao et al., 2020 [33]	China	8/8, RCT	E: BCI-FES + co-rehab MI task: KMI of wrist dorsiflexion C: co-rehab	Perform the MI task upon the appearance of the cue (“left” or “right”), the avatar would give the subjects visual feedback and the FES would be activated to cause the wrist dorsiflexion of the corresponding side.	FMA-UE	40 min/d, 3 d/wk, 4 wk	The average improvement score of the BCI group was 3.5, which was higher than that of the control group (0.9).
Cheng et al., 2020 [34]	Singapore	5/5, RCT	E: BCI-SRG + Soft Robotic Glove C: Soft Robotic Glove	Imagine ADL movement, like scanning goods, moving an object upward to a cabinet, etc.	FMA-UE, ARAT	120 min/d, 3 d/wk, 6 wk	No intergroup differences.
Chen et al., 2021 [35]	China	16/16, RCT	E: BCI-FES MI task: wrist-extension C: NMES	The electrode slices were attached to the skin above the two ends of the extensor carpi ulnaris	FMA-UE	40 min/d, 4 d/wk, 3 wk.	The FMA-UE was significantly higher than that in the sham group.

Abbr: Brain–computer interfaces, BCI; Functional Electrical Stimulation, FES; Neuromuscular Electrical Stimulation, NMES; the Fugl–Meyer Assessment Upper Extremity Scale, FMA-UE; Action Research Arm Test, ARAT; the Jebsen Hand Function Test, JHFT; the Goal Attainment Scale, GAS; Motor Activity Log, MAL; Modified Ashworth Scale, MAS; Electroencephalogram, EEG; Modified Barthel Index, MBI.

## Data Availability

The original contributions presented in the study are included in the article material, and further inquiries can be directed to the corresponding author.

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
