# Peer review of "Sensorimotor Rhythm-Based Brain–Computer Interfaces for Motor Tasks Used in Hand Upper Extremity Rehabilitation after Stroke: A Systematic Review"

_brainsci, 2022, doi:10.3390/brainsci13010056_

Round 1

Reviewer 1 Report

The paper presents the motor tasks adopted in SMR-based BCI, as well as the corresponding feedback adopted in the trials, and explored some articles in different libraries. The novelty  of this paper is not clear for me. However, I would like to present some comments to improve the article presentation

1. The summary needs to be reformulated

2. We need to see more comparisons with previous work to highlight the real contribution.

3. English still needs polishing. The manuscript should be better formatted and spelling and grammar should be carefully checked.

4. The paper is too short as a survey document, so please review the literature survey and include more results.

5. I propose to add  a discussion  to interpret and compare the results. It is adviced to be objective: point out the features and limitations of the work. 

6. Concerning the conclusion, let s say that the  purpose of the conclusion section is to restate the study aims and key questions and summarize the findings. In this paper, the conclusion is a repetion of what is already written in the abstract.

Author Response

Thank you very much for your time to review our manuscript.

Q1: “The summary needs to be reformulated”

Reply: Thank you for your suggestion. I have rewritten the summary. You can see this on page 2, lines 15-37.

Q2: “We need to see more comparisons with previous work to highlight the real contribution.”

Reply: Thank you for the valuable advice. We have added the content at line107-125.

“Clinical efficacy in hand function rehabilitation of stroke patients has been revealed by several reviews. Remsik et al.[21]considered BCIs as a method of hand function rehabilitation after stroke with a review. Monge-Pereira et al.[22] suggested EEG-based BCIs interventions may be a promising rehabilitation approach in subjects with stroke by a systematic review. Carvalho et al.[23] suggested that neurofeedback training with EEG-based BCIs might promote both clinical and neurophysiologic changes in stroke patients. Bai et al.[24] investigated the effectiveness of BCIs in restoring upper extremity function after stroke. Even though they have testified to the effectiveness of the BCIs, hardly have they forced deeply on these questions: why was a motor task chosen in each BCI trial, and is it suitable for a stroke patient? Some stroke patients received BCIs training but gained small improvements. Except for other reasons like a lack of treatment times, short treatment duration, etc., can the motor tasks help promote better motor recovery? Thus, this review concentrates on the motor tasks and feedback of BCI clinical trials based on upper limb and hand interventions with BCIs systems in patients after stroke, which is truly suitable for them to solve their problems. We use the traditional method to search articles and draw clinical recommendations. This review aims: (1) to explore the motor tasks design in EEG-based BCIs clinical trials, (2) to analyze the association between motor tasks and the neurologic mechanism, and (3) to discuss the feedback combing the motor tasks that were suitable for stroke patients.”

References:

  1. Remsik A, Young B, Vermilyea R, et al. A review of the progression and future implications of brain-computer interface therapies for restoration of distal upper extremity motor function after stroke. EXPERT REV MED DEVIC 2016;13:445-454
  2. Monge-Pereira E, Ibañez-Pereda J, Alguacil-Diego IM, et al. Use of Electroencephalography Brain-Computer Interface Systems as a Rehabilitative Approach for Upper Limb Function After a Stroke: A Systematic Review. PM&R 2017;9:918-932
  3. Carvalho R, Dias N, Cerqueira JJ. Brain‐machine interface of upper limb recovery in stroke patients rehabilitation: A systematic review. PHYSIOTHER RES INT 2019;24:e1764
  4. Bai Z, Fong KNK, Zhang JJ, Chan J, Ting KH. Immediate and long-term effects of BCI-based rehabilitation of the upper extremity after stroke: a systematic review and meta-analysis. J NEUROENG REHABIL 2020;17

Q3: “English still needs polishing. The manuscript should be better formatted and spelling and grammar should be carefully checked.”

Reply: Thank you very much. We have carefully polished our English.

Q4: “The paper is too short as a survey document, so please review the literature survey and include more results.”
Reply: Thank you so much for your advice. There were truly many other motor tasks for other diseases or healthy people, but it concentrated on some points, such as RCT, EEG-based BCI intervention, and evaluated by FMA-UE for stroke patients. Thus, after a series of screenings, only 15 articles were analyzed in this paper.

Q5: “I propose to add a discussion to interpret and compare the results. It is advised to be objective: point out the features and limitations of the work.”

Reply: We appreciate the advice. We have added a discussion to interpret and compare the results at lines 207-308, and the features and limitations of the work can be seen at line373-385.

Q6: “Concerning the conclusion, lets say that the purpose of the conclusion section is to restate the study aims and key questions and summarize the findings. In this paper, the conclusion is a repetion of what is already written in the abstract.”

Reply: Thank you for your valuable advice. I have rewritten the conclusion. “To sum up, we focus on the motor tasks adopted in EEG-based BCIs research, as well as the corresponding feedback adopted in the BCI trial from the very perspective of the clinic. Many motor tasks involve imagining or attempting to grasp or extend the hand and were matched with the BCIs-triggered robot or FES combined with visual feedback. To optimize BCI rehabilitation training, we should focus on patients’ difficulties during BCI training to help them to complete grasp motions, finger extension, thumb opposition, and other complex motions with the assistance of the robot or FES, or other combined feedback.”

Reviewer 2 Report

Dear Authors, 

thank you for giving me the opportunity to revise your manuscript entitled " Sensorimotor Rhythm-based Brain-computer Interfaces for Motor Tasks used in Hand upper extremity Rehabilitation After Stroke, a review". The manuscript investigated the effects of Brain computer interfaces in upper limb rehabilitation. The topic is very interesting and contribute to understand the real efficacy of a new technologic tool in rehabilitation. The paper is well written and succinct, nevertheless different critical issues should be addressed to make the paper suitable for publication:

Introduction: The introduction is well written, but should explain the main findings in new technological field in rehabilitation. BCI is a new tool in rehabilitation after stroke, but should be add in a framework of new technological devices, such as robotic rehabilitation. Moreover, disability related stroke should be more stressed in the introduction and not only limited on upper limb rehabilitation. I suggest to read these papers:  "Baricich A, Picelli A, Santamato A, Carda S, de Sire A, Smania N, Cisari C, Invernizzi M. Safety Profile of High-Dose Botulinum Toxin Type A in Post-Stroke Spasticity Treatment. Clin Drug Investig. 2018 Nov;38(11):991-1000. doi: 10.1007/s40261-018-0701-x.", "Calafiore D, Negrini F, Tottoli N, Ferraro F, Ozyemisci-Taskiran O, de Sire A. Efficacy of robotic exoskeleton for gait rehabilitation in patients with subacute stroke : a systematic review. Eur J Phys Rehabil Med. 2022 Feb;58(1):1-8. doi: 10.23736/S1973-9087.21.06846-5. and "de Sire A, Baricich A, Ferrillo M, Migliario M, Cisari C, Invernizzi M. Buccal hemineglect: is it useful to evaluate the differences between the two halves of the oral cavity for the multidisciplinary rehabilitative management of right brain stroke survivors? A cross-sectional study. Top Stroke Rehabil. 2020 Apr;27(3):208-214. doi: 10.1080/10749357.2019.1673592. "

Methods: Please, define the PICO of the research explain every analyzed outcome and the research methods, adding a table of the used keyword for each database . The authors should be specify the risk of bias of different studies ( ROB2 cochrane e.g.). Moreover, I suggest to use the JBI for RCT instead Pedro scale. Lastly, PROSPERO number should be add

Results: The results are quite confuse. First of all,, provide a paragraph and improve the table with detailed study characteristics (years, country, assessment time for each study). I suggest to list each included study and explain the exact experimental design. Move the paragraph assessment in methods

Discussion: In the introduction the authors have cited different review about BCI and post-stroke rehabilitation. Which are the novelties of your work? Moreover, the authors should be explain the limitation of the study. 

Best regards

Author Response

Thank you very much for your time to review our manuscript.

Introduction: The introduction is well written, but should explain the main findings in new technological field in rehabilitation. BCI is a new tool in rehabilitation after stroke, but should be add in a framework of new technological devices, such as robotic rehabilitation. Moreover, disability related stroke should be more stressed in the introduction and not only limited on upper limb rehabilitation. I suggest to read these papers:  "Baricich A, Picelli A, Santamato A, Carda S, de Sire A, Smania N, Cisari C, Invernizzi M. Safety Profile of High-Dose Botulinum Toxin Type A in Post-Stroke Spasticity Treatment. Clin Drug Investig. 2018 Nov;38(11):991-1000. doi: 10.1007/s40261-018-0701-x.", "Calafiore D, Negrini F, Tottoli N, Ferraro F, Ozyemisci-Taskiran O, de Sire A. Efficacy of robotic exoskeleton for gait rehabilitation in patients with subacute stroke : a systematic review. Eur J Phys Rehabil Med. 2022 Feb;58(1):1-8. doi: 10.23736/S1973-9087.21.06846-5. and "de Sire A, Baricich A, Ferrillo M, Migliario M, Cisari C, Invernizzi M. Buccal hemineglect: is it useful to evaluate the differences between the two halves of the oral cavity for the multidisciplinary rehabilitative management of right brain stroke survivors? A cross-sectional study. Top Stroke Rehabil. 2020 Apr;27(3):208-214. doi: 10.1080/10749357.2019.1673592. "

Reply: Thank you so much for the advice. I have added some new technological fields in rehabilitation, ‘Various methods were being applied in hand function rehabilitation, such as central interventions like mirror therapy, transcranial magnetic stimulation, transcranial direct current stimulation, brain-computer interfaces (BCIs), motor imagery, etc., peripheral interventions like a robot, physical therapy, functional electrical stimulation, etc., and medicine like botulinum for spasticity[5]’, and cite ‘Baricich A, Picelli A, Santamato A, Carda S, de Sire A, Smania N, Cisari C, Invernizzi M. Safety Profile of High-Dose Botulinum Toxin Type A in Post-Stroke Spasticity Treatment. Clin Drug Investig. 2018 Nov;38(11):991-1000. doi: 10.1007/s40261-018-0701-x’ as a means for hand recovery. And of course, disability-related stroke is also stressed in the introduction. ‘Stroke causes the highest morbidity associated with disability-adjusted life years lost in China, with two million new cases annually[1]. Up to 66% of stroke survivors experience upper limb and hand motor impairments, which results in functional limitations in activities of daily living and decreased life quality[2,3], and leads to a heavy burden for the family and society. Hand rehabilitation after a stroke is difficult during neurorehabilitation.’

  1. Kwah LK, Harvey LA, Diong J, Herbert RD. Models containing age and NIHSS predict recovery of ambulation and upper limb function six months after stroke: an observational study. J PHYSIOTHER 2013;59:189-197
  2. Morris JH, van Wijck F, Joice S, Donaghy M. Predicting health related quality of life 6 months after stroke: the role of anxiety and upper limb dysfunction. DISABIL REHABIL 2013;35:291-299

Methods: Please, define the PICO of the research explain every analyzed outcome and the research methods, adding a table of the used keyword for each database. The authors should be specify the risk of bias of different studies ( ROB2 cochrane e.g.). Moreover, I suggest to use the JBI for RCT instead Pedro scale. Lastly, PROSPERO number should be add.

1.define the PICO of the research explain every analyzed outcome and the research methods

Reply: Thank you so much for the advice. I have defined the PICO for the research. Line 139-149.

  1. A table of the used keyword for each database

Reply: The content is really huge, so I put it in a .doc as a supplementary file, and it can be found in supplement 1. I also referred to the supplement in the paper at l ine136.

  1. I suggest to use the JBI for RCT instead Pedro scale.

Reply: And about the PEDro for articles quality check. I appreciate the JBI you’ve mentioned. As the reason I chose PEDro, I referred to the similar BCI-related literature “BAI Z, FONG K N K, ZHANG J J, et al. Immediate and long-term effects of BCI-based rehabilitation of the upper extremity after stroke: a systematic review and meta-analysis[J]. Journal of NeuroEngineering and Rehabilitation, 2020,17(1)”, “BANIQUED P D E, STANYER E C, AWAIS M, et al. Brain–computer interface robotics for hand rehabilitation after stroke: a systematic review[J]. Journal of NeuroEngineering and Rehabilitation, 2021,18(1).”, “CARVALHO R, DIAS N, CERQUEIRA J J. Brain‐machine interface of upper limb recovery in stroke patients rehabilitation: A systematic review[J]. Physiotherapy Research International, 2019,24(2):e1764.”, they all used PEDro to qualify the article.

  1. specify the risk of bias of different studies ( ROB2 cochrane e.g.)

Reply: We have already used PEDro for quality checks. thus, concerning our review type. Maybe we do not need the risk of bias in different studies. Thank you for your valuable advice. We will apply ROB2 Cochrane in our further new paper.

5.‘PROSPERO number’

Reply: Thank you for your advice. We found it is usually registered for meta-analysis, and our paper was not a meta-analysis, so we didn’t register on the PROSPERO site.

Results:

  1. provide a paragraph and improve the table with detailed study characteristics (years, country, assessment time for each study)

Reply: Thank you for your advice. We have added table 1 with the detailed study characteristics (years, country, assessment time for each study), lines 271-272;

  1. list each included study and explain the exact experimental design

Reply: Thank you. I have also added the information to Table 2, at lines 271-272;

3.“Move the paragraph assessment in methods”,

Reply: Thank you for your valuable advice, I have moved it to the methods part. Lines 145;

  1. Which are the novelties of your work?

Reply: Thank you for your advice. we focus on the motor tasks adopted in EEG-based BCIs research, as well as the corresponding feedback adopted in the BCI trial from the very perspective of the clinic “This review aims: (1) to explore the motor tasks design in EEG-based BCIs clinical trials, (2) to analyze the association between motor tasks and the neurologic mechanism, and (3) to discuss the feedback combing the motor tasks that were suitable for stroke patients.”

  1. explain the limitation of the study.

Reply: Thank you for the advice. I have added this part in lines 379-382.

“However, there are some limitations to this review. First, our review only focuses on RCT. Thus, more motor tasks can’t be presented. Second, the effects of different motor tasks and different feedback were not quantified.”

Round 2

Reviewer 2 Report

Dear Authors,

At the light of the outstanding revision process, the paper is suitable for fully publication in journal.

Best Reagards